# Abdominal Expansion versus Abdominal Drawing-In Strategy on Thickness and Electromyography of Lumbar Stabilizers in People with Nonspecific Low Back Pain: A Cross-Sectional Study

**DOI:** 10.3390/ijerph18094487

**Published:** 2021-04-23

**Authors:** Yi-Liang Kuo, Chieh-Yu Kao, Yi-Ju Tsai

**Affiliations:** 1Department of Physical Therapy, College of Medicine, National Cheng Kung University, Tainan 701, Taiwan; yiliangkuo@mail.ncku.edu.tw; 2Department of Rehabilitation, Sengkang Community Hospital 1 Anchorvale Street, Singapore 544835, Singapore; jeiyukao@gmail.com; 3Institute of Allied Health Sciences, College of Medicine, National Cheng Kung University, Tainan 701, Taiwan

**Keywords:** low back pain, ultrasonography, muscle contraction, abdominal muscles, lumbosacral region

## Abstract

The abdominal expansion (AE) strategy, involving eccentric contraction of the abdominal muscles, has been increasingly used in clinical practices; however, its effects have not been rigorously investigated. This study aimed to investigate the immediate effects of the AE versus abdominal drawing-in (AD) strategy on lumbar stabilization muscles in people with nonspecific low back pain (LBP). Thirty adults with nonspecific LBP performed the AE, AD, and natural breathing (NB) strategies in three different body positions. Ultrasonography and surface electromyography (EMG) were, respectively, used to measure the thickness and activity of the lumbar multifidus and lateral abdominal wall muscles. The AE and AD strategies showed similar effects, producing higher EMG activity in the lumbar multifidus and lateral abdominal wall muscles when compared with the NB strategy. All muscles showed higher EMG activity in the quiet and single leg standing positions than in the lying position. Although the AE and AD strategies had similar effects on the thickness change of the lumbar multifidus muscle, the results of thickness changes of the lateral abdominal muscles were relatively inconsistent. The AE strategy may be used as an alternative method to facilitate co-contraction of lumbar stabilization muscles and improve spinal stability in people with nonspecific LBP.

## 1. Introduction

Spinal instability is an important cause of lower back pain (LBP) [1,2]. The abdominopelvic cavity is surrounded by the diaphragm superiorly, transverses abdominis (TrA) anteriorly, lumbar multifidus (MF) posteriorly, and pelvic floor muscles inferiorly. Once activated, these deep muscles can provide stability to the spine by increasing intra-abdominal pressure [3]. Previous studies observed atrophy, fatty infiltration, and neural inhibition of the TrA and MF muscles among people with LBP [4,5]. They also reported reduced mobility of the diaphragm and reduced respiratory muscle endurance [6]. These changes have a destabilizing influence on the spine and may result in chronic and/or recurrent symptoms in people with LBP.

The abdominal drawing-in (AD) strategy is one of the commonly used volitional pre-emptive abdominal contraction (VPAC) strategies among rehabilitation professionals. The AD strategy involves concentric contraction of the abdominal muscles. Previous studies reported that the AD strategy improve control of deep lumbar stabilization muscles [7]. Ultrasound imaging (USI) studies verified that the AD strategy is effective in increasing the thickness of the TrA and MF in healthy adults and patients with LBP [8,9,10]. After the AD strategy was implemented in different body positions and in combination with various dynamic exercises, patients with LBP showed positive outcomes regarding self-reported pain and physical disability [11,12].

The abdominal expansion (AE) strategy has been increasingly used by a therapeutic approach, dynamic neuromuscular stabilization, prior to purposeful movement during daily activities, and as the essential foundation of their stabilization exercises [13,14,15]. In contrast to the AD strategy, the AE strategy involves eccentric contraction of the abdominal muscles accompanied by downward contraction of the diaphragm. The co-contraction of these muscles maintains the dome shape of the diaphragm and facilitates inspiration, which is theorized to assist spinal stability by modulating intra-abdominal pressure [16]. Previous studies reported that patients with cerebrovascular accident practicing the AE strategy demonstrated improved functional reach distance and reduced center of pressure displacement and sway velocity in unsupported sitting [17,18]. However, the AE strategy may have different effects in patients with cerebrovascular accident and those with LBP.

To date, only one study has investigated the effect of the AE strategy on lumbar stabilization muscles in patients with LBP. Surface electromyography (EMG) data revealed that the AE strategy was effective in increasing internal abdominal oblique (IO) and MF muscle activities in the forward kneeling and supine bridging positions in patients with lumbar spinal instability [19]. That study was published in Korean, and limited information could be extracted from the English abstract. Moreover, that study was conducted in a specific subgroup of patients with LBP. Further research is needed to clarify the effects of the AE strategy in people with nonspecific LBP.

Therefore, through both USI and EMG measurements, this study investigated the immediate effect of the AE strategy on lumbar stabilization muscles in people with LBP. We hypothesized that the AE strategy is more effective than no VPAC strategy (natural breathing, NB) and is as effective as the AD strategy in activating deep lumbar stabilization muscles. We also examined the effects of the AE strategy in different postural conditions because postural demand may influence muscle contractions. The strict social distancing and home confinement as a result of the current COVID-19 situation greatly influenced how patients with musculoskeletal disorders are cared around the world [20]. Therapeutic exercise, such as the AE strategy that does not require close proximity, may be an appropriate treatment approach for some patients with LBP.

## 2. Materials and Methods

### 2.1. Study Design

This was a cross-sectional study investigating the immediate effects of VPAC strategies (NB, AD, and AE) performed in different body positions (lying, quiet standing, and single leg standing) on the activation of lumbar stabilization muscles in people with nonspecific LBP. USI and EMG were used to, respectively, measure the thickness and activity of the MF and lateral abdominal wall muscles (TrA, IO, and external abdominal oblique [EO] muscles). This study was approved by the Institutional Ethical Review Board (Approval No. B-ER-106-372).

### 2.2. Participants

A priori power analysis indicated that a sample size of 30 was needed (moderate effect size, 0.5; α, 0.025; power, 0.95) [21]. A convenience sample of adults with nonspecific LBP was recruited from the neighboring rehabilitation and physical therapy clinics through flyers (Figure 1). Equal numbers of male and female participants were recruited to minimize the gender bias. All participants provided written informed consent before data collection.

The inclusion criteria were as follows: (1) age between 20 and 50 years; (2) unilateral or bilateral pain between the 12th thoracic vertebra and the gluteal line; (3) current pain intensity ≥ 2 as per the numeric rating scale The exclusion criteria were as follows: (1) current neurological symptoms or signs related to the back; (2) specific back-related conditions (e.g., spinal deformities, fractures or surgery); (3) presence of abdominal, cardiopulmonary, or gastrointestinal conditions that interfere with daily living within the past 6 months; (4) urinary conditions; (5) medical conditions that affect balance (e.g., stroke, vestibular disorders, and cancer); (6) current participation in structured and supervised core or trunk exercise training programs; (7) body mass index of >30 kg/m^2^ [22]; and (8) pregnancy.

### 2.3. Procedure

Participants provided basic data, pain location and intensity, disease-specific disability, and fear-avoidance belief. After the practice of VPAC strategies following standardized verbal instructions, both USI and EMG measurements were performed for the NB strategy (simple inhalation and exhalation as usual). Then, participants performed the AE and AD strategies in a random order. For the AD strategy, participants drew the navel up and in toward the spine and held the contraction during breathing. For the AE strategy, participants slowly pushed their lower abdomen outward as in diaphragmatic inhalation and then held the contraction during breathing. All participants were instructed not to deeply inhale or exhale and keep the spine and pelvis steady for all strategies applied. A ten-minute rest was given between each VPAC strategy. Measurements were taken in the lying position as the baseline condition, and the other two positions were assumed randomly to reduce the effect of learning or fatigue.

The Siemens ultrasound system (ACUSON NX3TM, Siemens Medical Solution Inc., Issaquah, WA, USA) with a linear transducer of 4–12 MHz (VF12-4) set in B mode was used to measure muscle thickness. The Delsys wireless EMG system (Trigno^TM^ system, Delsys Inc., Boston, MA, USA) with a sampling rate of 2000 Hz was used to measure muscle activity. Because USI measurement prevented the placement of surface EMG electrodes for simultaneous USI and EMG measurement of the same muscle, the USI transducer was positioned on the targeted muscles of the symptomatic side based on the self-report of each participant, and EMG electrodes were attached to the opposite side of the muscle bellies.

For measurement of the lateral abdominal wall, the USI transducer was placed perpendicular to the mid-axillary line approximately 10 cm lateral to the umbilicus first, and then was adjusted until the anterior edge of the TrA became visible on the screen (Figure 2) [23,24]. For measurement of the MF muscle, the USI transducer was aligned longitudinally along the lumbar spine at the L5-S1 level, and was then moved laterally and angled slightly medially (Figure 2). The same investigator (C. Y. Kao) took all of the USI images of all participants in order to minimize measurement errors during image acquisition.

Surface EMG electrodes were attached on the EO (at the midpoint between the lowest part of the ribcage and iliac crest) [25], IO/TrA (2 cm medial and inferior from the ASIS) [26], and MF (1 cm lateral from the L5 spinous process) [21]. During testing, participants were instructed to maintain each VPAC strategy for 5 s, and ultrasound images taken at the end of expiration, as determined by visual inspection of the abdomen. Simultaneously, 5-s of EMG data were recorded. Two testing trials were conducted for each condition, with a 30-s interval between trials. Average data of two trials were used for data analysis.

### 2.4. Data Reduction

Muscle thickness was measured offline using ImageJ (National Institutes of Health, Bethesda, MD, USA). MF thickness was measured from the echogenic tip of the L5-S1 zygapophyseal joint to the superficial fascia of the muscle (Figure 2). The thickness of each lateral abdominal wall muscle was measured from the superior to the inferior fascial borders at the thickest part of the muscle (Figure 2).

The EMG data from the middle 3 s of the 5-s data trials were processed and analyzed using the computer algorithms written in the Matlab language (R2018b, The MathWorks Inc., Natick, MA, USA). All EMG raw signals were first filtered using a band pass filter of 20–300 Hz with the fourth-order Butterworth filter to reduce movement artifacts first and were then filtered using a band stop filter of 59.5–60.5 Hz to eliminate the powerline interference. The filtered EMG data were rectified and processed with a moving average of 100 ms based on the equation: $y\left(t\right)=\frac{1}{T}\int_{t-T}^{t}x\left(t\right)dt$, where *T* = 100 ms. The integrated EMG values of each muscle were further calculated.

### 2.5. Statistical Analysis

All data are presented as mean and standard deviation (SD). Two-way analysis of variance (ANOVA) for repeated measures (3 VPAC strategies × 3 positions) was performed for each dependent variable. One-way repeated measures ANOVA was performed to determine differences among VPAC strategies at each position if the interaction effect was significant. Statistical significance for repeated measures ANOVA was set at *p* < 0.05, and the Bonferroni-corrected significance level was used for multiple comparisons. SPSS statistics for Windows (Version 17.0, SPSS Inc., Chicago, IL, USA) was used for statistical analyses.

## 3. Results

Fifteen men and 15 women with nonspecific LBP participated in this study. Table 1 presents their characteristics, and Figure 3 and Figure 4, respectively, illustrate the mean thickness and EMG activity of lumbar stabilization muscles during different VPAC strategies and positions. The thickness and EMG activity of the MF muscle were significantly affected by the VPAC and position main effects, but not by their interaction effects (Table 2). Both the AE and AD strategies significantly increased the thickness and EMG activity of the MF muscle compared with the NB strategy. For the AE and AD, the mean difference in MF muscle thickness was 2.0 mm (*p* < 0.001) and 1.2 mm (*p* < 0.001), and its mean difference in EMG activity was 18.1 mv (*p* = 0.015) and 10.8 mv (*p* = 0.001), respectively (Table 3). No significant differences were observed in MF muscle thickness and EMG activity during the AE and AD strategies (Table 3).

TrA muscle thickness was significantly affected by the interaction of the VPAC strategy and body position (Table 2). When analyzing the simple effect of the VPAC strategy by position, the AD strategy significantly increased TrA muscle thickness in all three positions when compared with the NB strategy (Table 2). The mean difference in TrA muscle thickness for the AD and NB strategies ranged between 1.4 mm and 2.0 mm (Table 3). TrA muscle thickness during the AE and NB strategies was not significantly different (Table 3). IO muscle thickness was significantly affected by the VPAC and position main effects but not by their interaction effects (Table 2). IO muscle thickness significantly increased when using the AD strategy compared with when using the NB strategy. The mean difference in IO muscle thickness between the AD and NB strategies was 1.8 mm (*p* < 0.001, Table 3). No significant differences were observed in IO muscle thickness during the AE and NB strategies (*p* > 0.999, Table 3).

The EMG activity of the IO/TrA muscles was significantly affected by the VPAC and position main effects, but not by their interaction effects (Table 2). The EMG activity of the IO/TrA muscles was the highest during the AD strategy, followed by the AE strategy and the NB strategy (Figure 4). The EMG activity of the IO/TrA muscles during all three VPAC strategies was significantly different (Table 3).

EO muscle thickness was significantly affected only by the VPAC main effect but not by the position main effect and interaction effects (Table 2). Compared with the NB strategy, EO muscle thickness significantly increased using the AD strategy but decreased significantly during the AE strategy (Table 2). The mean thickness difference in the EO muscle was 0.5 mm between the AD and NB strategies (*p* = 0.036) and 0.8 mm between the AE and NB strategies (*p* < 0.001) (Table 3). Conversely, the EMG activity of the EO muscle was significantly affected by the VPAC strategy and position main effects and their interactions (Table 2). Both AE and AD strategies significantly increased the EMG activity of the EO muscle compared with the NB strategy. The mean difference in EMG activity of the EO muscle ranged from 11.1 mv to 21.9 mv between the AE and NB strategies and from 16.3 mv to 22.7 mv between the AD and NB strategies (Table 3). The EMG activity values of the EO muscle during the AE and AD strategies were not significantly different (Table 3).

Overall, the measured muscles showed higher EMG activity in the quiet and single leg standing positions than in the lying position (Table 2). A similar result was obtained for the thickness change of the MF muscle, that is, significantly thicker in quiet and single leg standing positions than in the lying position (Table 2). However, the results of the effect of body position on the thickness of the lateral abdominal wall muscles were inconsistent (Table 2).

## 4. Discussion

Improving spinal stability has been a focus for prevention and treatment for LBP [8,10,21,27]. To the best of our knowledge, this is the first study to investigate the immediate effect of the AE strategy on lumbar stabilization muscles in people with nonspecific LBP using both USI and EMG measurements. Our results demonstrated that the AE strategy had effects similar to the AD strategy in people with nonspecific LBP, producing higher EMG activity in the MF and lateral abdominal wall muscles compared with the NB strategy. The AE strategy also had a similar effect as the AD strategy on thickness change of the MF muscle. Based on these results, the AE strategy has potential to be used as an alternative method to facilitate co-contraction of lumbar stabilization muscles in people with LBP, although the results of its effect on the thickness change of the lateral abdominal wall muscles were relatively inconsistent.

Most of our results support our hypothesis that the AE strategy is more effective than the NB strategy in activating deep lumbar stabilization muscles. The EMG activity of the MF and IO/TrA muscles during the AE strategy was significantly higher than that during the NB strategy. Compared with the NB strategy, the AE strategy also significantly increased the thickness of the MF muscle but not of the TrA muscle. There is a growing trend to use the USI to assess muscle functions [28]. An increase in muscle thickness measured with USI is often interpreted as an increase in muscle activity during concentric contraction because muscle thickness measured with USI is associated with muscle activity measured with EMG [29]. Previous studies have proved the intra-rater reliability of USI measurement in healthy people and in patients with LBP [23,29]. Our study did not show consistent results between the EMG and USI data, especially for the TrA muscle. The abdomen is pushed outward with abdominal breathing during the AE strategy; the thickness of the lateral abdominal wall muscles is expected to decrease with eccentric contraction. However, factors such as the extensibility of a musculotendinous unit and the contraction of adjacent muscles might influence USI measurement of muscle thickness [28], especially for long and slender lateral abdominal wall muscles. Considering that significantly increased EMG activity was observed in all lateral abdominal wall muscles during the AE strategy, the results of nonsignificant thickness change in lateral abdominal wall muscles, as measured using USI, when compared with the NB strategy should be interpreted with caution.

The results of significant position main effects on the EMG activity of the lumbar stabilization muscles were not surprising. Standing positions place more postural demand on lumbar stabilization muscles compared with the lying position. A previous study revealed a positive correlation between TrA muscle activation and postural demand in the standing position by varying the height of bilateral arm positions while holding a 3 kg dumbbell in each hand [30]. Our results are consistent with those of previous studies [19,23], demonstrating that despite LBP, participants have the ability to voluntarily engage in the AD and AE strategies in a more challenging posture. The results of the position main effect on the thickness of lumbar stabilization muscles were inconsistent with EMG measurements. One possible explanation is the limitation of USI measurements as discussed earlier.

The majority of previous studies focus on the TrA muscle when investigating the effect of therapeutic exercise to improve spinal stability. The MF muscle is an equally important stabilizer of the lumbar spine [31]. Localized atrophy of the MF muscle is reported to be strongly associated with LBP [32,33]; however, the voluntary contraction of the MF muscle through exercise is extremely difficult. The common method is to instruct patients to gently swell out or contract the MF under the therapist’s palpating finger without moving the spine or pelvis [34]. Clinically, individuals who have been educated or informed about the location of the MF muscle may not achieve a voluntary isometric contraction despite practice. Previous studies reported no significant difference in EMG activity or cross-sectional area of the MF muscle using magnetic resonance imaging when attempts were made to activate it using various trunk or leg extension movements [35,36]. The AE strategy used in this study can be easily comprehended and performed compared with the abstruse instruction of swelling out the MF muscle. The effects of the AE strategy in patients with LBP should be further investigated through longitudinal intervention studies.

Our study has several limitations. First, USI and EMG measurements of the MF and lateral abdominal wall muscles were not performed on the same side. Therefore, the interpretation of the results may be influenced by side-to-side differences. Second, the diaphragm contributes to spinal stability and is considerably involved during the AE strategy. However, USI and EMG measurements of the diaphragm were not performed because of the concern of the invasive procedure and time required. Consequently, the mechanism of the AE strategy cannot be completely understood. Considering that the overall prevalence of LBP is higher in women than in men [37], equal numbers of male and female participants were recruited in this study. However, the sample for both genders was too small to infer any gender difference. Finally, whether the changes observed in the thickness and EMG activity of the muscles induced by the AE strategy will transfer to improvements in pain and physical functions is unclear. Future research should investigate these outcomes in patients with LBP and consider measurement of the diaphragm.

## 5. Conclusions

The AE and AD strategies have similar effects in people with nonspecific LBP, producing larger thickness change and higher EMG activity in deep lumbar stabilization muscles compared with the NB strategy. While some individuals have the difficulty performing the traditional AD strategy, the AE strategy may be used as an alternative method to facilitate co-contraction of the lumbar stabilization muscles and improve spinal stability in people with LBP.

## Figures and Tables

**Figure 1 ijerph-18-04487-f001:**
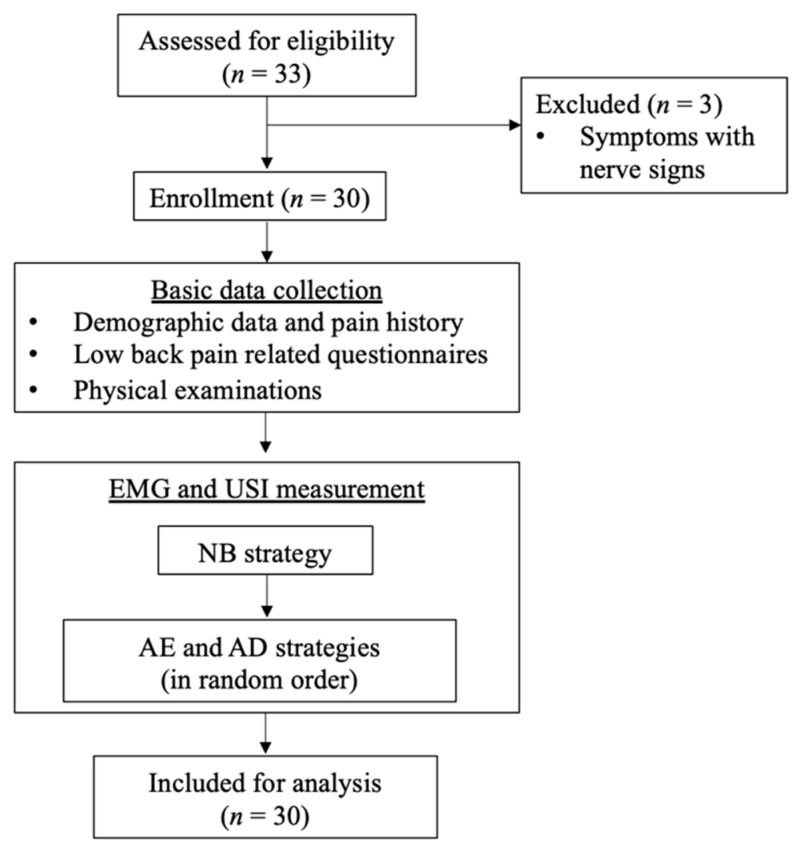
Flowchart of participant recruitment and data collection. EMG: electromyography; USI: ultrasound imaging; NB: natural breathing; AD: abdominal drawing-in; AE: abdominal expansion.

**Figure 2 ijerph-18-04487-f002:**
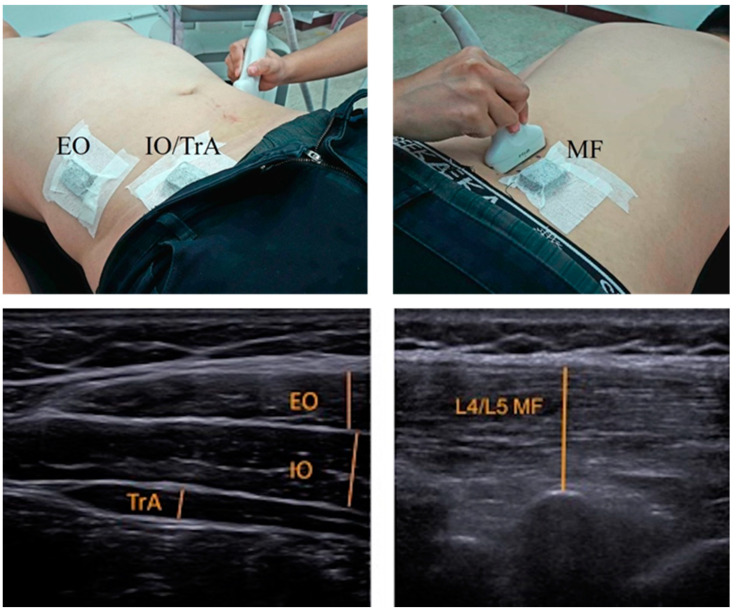
Positions of the ultrasound transducer and surface electromyography sensor (**top**) and ultrasound images of the lumbar multifidus and lateral abdominal wall muscles (**bottom**). EO: external abdominal oblique; IO: internal abdominal oblique; TrA: transverse abdominis; MF: lumbar multifidus.

**Figure 3 ijerph-18-04487-f003:**
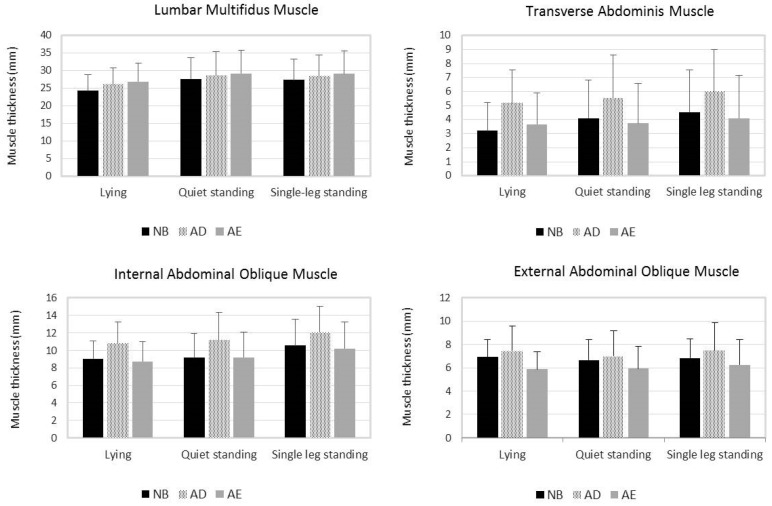
Mean thickness of the lumbar stabilization muscles during natural breathing (NB), abdominal drawing-in (AD), and abdominal expansion (AE) strategies in different body positions. Error bars indicate standard deviations.

**Figure 4 ijerph-18-04487-f004:**
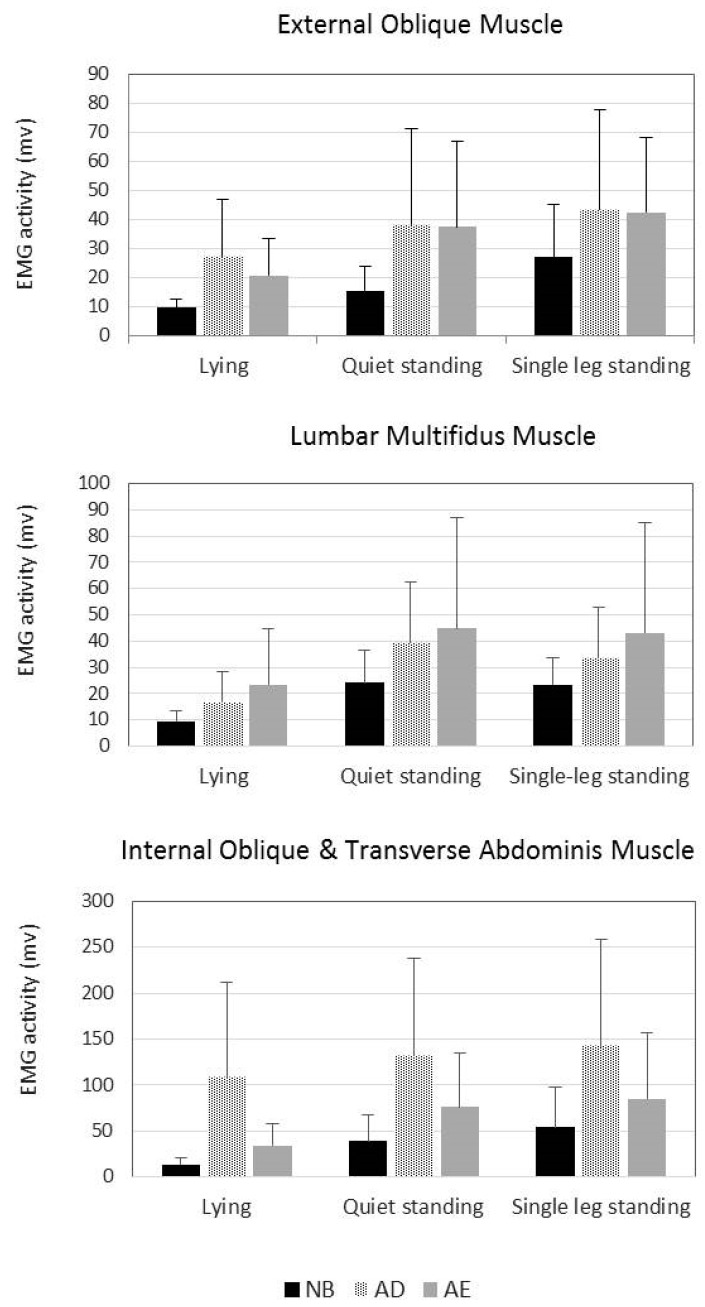
Mean electromyography activity of the lumbar stabilization muscles during natural breathing (NB), abdominal drawing-in (AD), and abdominal expansion (AE) strategies in different body positions. Error bars indicate standard deviations.

**Table 1 ijerph-18-04487-t001:** Participant baseline characteristics (*n* = 30).

Variable	Total *n* = 30
Age (year)	26.7 ± 7.0
Height (cm)	167.0 ± 8.6
Weight (kg)	63.3 ± 9.4
Body mass index (kg/m2)	22.7 ± 2.5
Pain distribution (number)	
Unilateral	17
Bilateral	13
Numeric rating scale (points/10)	3.4 ± 1.4
Fear-avoidance belief questionnaire	
Physical (points/24)	19.9 ± 4.0
Work (points/42)	21.5 ± 9.1
Oswestry disability index (%)	11.7 ± 7.0

**Table 2 ijerph-18-04487-t002:** Comparisons for the thickness and activity of the lumbar stabilization muscles between different volitional preemptive abdominal contraction (VPAC) strategies and body positions.

Variable	VPAC			Position			Interaction	
	F	*p*	Post-hoc	F	*p*	Post-hoc	F	*p*
Muscle thickness								
MF	19.72	<0.001	AE, AD > NB	20.07	0.001	SLS, QTS > LYI	1.94	0.147
TrA	63.22	<0.001	LYI: AD > AE, NB QTS: AD > AE, NB SLS: AD > AE, NB	17.61	<0.001	NB: SLS, QTS > LYI AD: SLS > QTS, LYI AE: NS	4.75	0.004
IO	28.78	<0.001	AD > AE, NB	12.44	<0.001	SLS > QTS, LYI	0.63	0.591
EO	24.07	<0.001	AD > NB > AE	1.70	0.200	NS	1.87	0.120
Muscle activity								
MF	6.16	0.012	AE, AD > NB	36.57	<0.001	SLS, QTS > LYI	1.01	0.375
IO/TrA	20.97	<0.001	AD > AE > NB	15.29	<0.001	SLS, QTS > LYI	1.02	0.363
EO	12.79	<0.001	LYI: AE, AD > NB QTS: AE, AD > NB SLS: AE, AD > NB	20.86	<0.001	NB: SLS > QTS > LYI AD: SLS, QTS > LYI AE: SLS, QTS > LYI	2.54	0.044

Abbreviation: VPAC = volitional preemptive abdominal contraction; MF = lumbar multifidus; TrA = transverse abdominis; EO = external abdominal oblique; IO = internal abdominal oblique; AE = abdominal expansion strategy; AD = abdominal drawing-in strategy; NB = natural breathing strategy; LYI = lying; QTS = quiet standing; SLS = single leg standing; NS: non-significant.

**Table 3 ijerph-18-04487-t003:** Comparisons for the thickness and activity of the lumbar multifidus and lateral abdominal wall muscles between different volitional preemptive abdominal contraction (VPAC) strategies.

Muscle		AE versus AD			AE versus NB			AD versus NB		
		∆	95% CI	*p*	∆	95% CI	*p*	∆	95% CI	*p*
Muscle thickness										
MF ^1^	Overall	0.7	−0.2, 1.7	0.193	2.0	1.2, 2.7	<0.001	1.2	0.6, 1.9	<0.001
TrA ^2^	LYI	−1.5	−2.0, −1.0	<0.001	0.4	−0.1, 0.9	0.091	2.0	1.5, 2.4	<0.001
	QTS	−1.7	−2.4, −1.1	<0.001	−0.4	−0.8, 0.1	0.203	1.4	0.9, 1.9	<0.001
	SLS	−1.9	−2.7, −1.1	<0.001	−0.4	−1.0, 0.1	0.178	1.5	1.0, 1.9	<0.001
IO	Overall	−2.0	−2.9, −1.1	<0.001	−0.2	−0.9, 0.5	>0.999	1.8	1.3, 2.3	<0.001
EO	Overall	−1.3	−1.9, −0.8	<0.001	−0.8	−1.2, −0.4	<0.001	0.5	0.03, 1.0	0.036
Muscle activity										
MF	Overall	7.3	−8.5, 23.0	0.754	18.1	2.9, 33.2	0.015	10.8	4.0, 17.7	0.001
IO/TrA	Overall	−61.9	−101.9, −21.8	0.002	29.5	9.0, 50.0	0.003	91.3	46.5, 136.1	<0.001
EO	LYI	−6.6	−13.9, 0.74	0.090	11.1	5.5, 16.8	<0.001	17.7	8.5, 27.0	<0.001
	QTS	−0.9	−10.9, 9.2	>0.999	21.9	8.1, 35.6	0.001	22.7	7.7, 37.8	0.002
	SLS	−0.9	−12.1, 10.3	>0.999	15.4	3.7, 27.0	0.007	16.3	0.1, 32.4	0.048

Abbreviation: MF = lumbar multifidus; TrA = transverse abdominis; EO = external abdominal oblique; IO = internal abdominal oblique; LYI = lying; QTS = quiet standing; SLS = single leg standing; AE = abdominal expansion strategy; AD = abdominal drawing-in strategy; NB = natural breathing strategy; CI = confidence interval; ∆ = mean difference. ^1^ The interaction effect was not significant; therefore, the VPAC effect was analyzed regardless of the position. ^2^ The interaction effect was significant; therefore, the VPAC effect was analyzed at each position.

## Data Availability

The data presented in this study are available on request from the corresponding author.

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
