# Peer review of "Abdominal Expansion versus Abdominal Drawing-In Strategy on Thickness and Electromyography of Lumbar Stabilizers in People with Nonspecific Low Back Pain: A Cross-Sectional Study"

_ijerph, 2021, doi:10.3390/ijerph18094487_

Round 1

Reviewer 1 Report

Very clearly written manuscript. There are a few minor comments.

Line 58 Re-phase sentence so that "stroke practicing" are not adjacent as at first read it came across as individuals were practicing a physical exercise

Line 139-Was the mean of the two testing trials taken, or just the better of the two testing trials used in the analysis. Please clarify.

Line 251-rephase without the word "nowadays", as while perfectly fine for less formal oral speech, however in written form it strikes a wrong tone 

Since there were 15 men and 15 women, and this might not be adequately powered to determine sex differences, but were there trends male vs female. This could be addressed in the Discussion section, future direction or limitation.

Reviewer 2 Report

Dear Authors,

Congratulations on the work done: it is well justified, the methodology has been adequately described and the arguments developed in Discussion are solid and adequate.

Perhaps, to enrich the manuscript, it would be of interest that you take into account that:

1. The Introduction and / or Discussion could be enriched if the topic addressed is contextualized with the current situation of strict and moderate confinements that is being experienced throughout the world (and its consequent impact on physical and mental health):

Rodríguez-Nogueira Ó, Leirós-Rodríguez R, Benítez-Andrades JA, Álvarez-Álvarez MJ, Marqués-Sánchez P, Pinto-Carral A. Musculoskeletal pain and teleworking in times of the COVID-19: Analysis of the impact on the workers at two spanish universities. Int J Environ Res Public Health. 2021; 18: 31-42. Doi: 10.3390 / ijerph18010031

Leirós-Rodríguez R, Rodríguez-Nogueira Ó, Pinto-Carral A, Álvarez-Álvarez MJ, Galán-Martín MÁ, Montero-Cuadrado F, Benítez-Andrades JA. Musculoskeletal pain and non-classroom teaching in times of the COVID-19 pandemic: Analysis of the impact on students from two Spanish universities. J Clin Med. 2020; 9: 4053-4064. doi: 10.3390 / jcm9124053.

2. It is incorrect from a formal point of view to include zero as the last decimal place (the zeros to the right of the decimal point do not mean anything). 

Kind regards.

Reviewer 3 Report

Authors evaluated the effects of the abdominal expansion maneuver in people with back pain. Some issues to be solved:

  1. The originality of the paper should be stressed. Actually, english-based papers can be found in literature (https://doi.org/10.18857/jkpt.2015.27.3.147, https://doi.org/10.14474/ptrs.2016.5.3.113); for this reason authors must necessarly provide an exact overview of the literature making easier for the readers to understand the scientific impact of the outcomes and the novelties.
  2. Methods lack of information on Ethical Committee approval
  3. More details on electrode placement are needed. Which guidelines were followed?
  4. Was the operator performing the ultrasonic measurement always the same?
  5. How did authors compute the EMG activity? Please better explain through equation
  6. Authors should stress the advantages of the technique in clinical routines

Round 2

Reviewer 3 Report

Authors answered to all my previous concerns.